# Intra-Specific Variation in Desiccation Tolerance of *Citrus sinensis ‘bingtangcheng’* (L.) Seeds under Different Environmental Conditions in China

**DOI:** 10.3390/ijms24087393

**Published:** 2023-04-17

**Authors:** Hongying Chen, Anne M. Visscher, Qin Ai, Lan Yang, Hugh W. Pritchard, Weiqi Li

**Affiliations:** 1Germplasm Bank of Wild Species, Kunming Institute of Botany, Chinese Academy of Sciences, Kunming 650201, China; 2Trait Diversity and Function Department, Royal Botanic Gardens, Kew, Wakehurst, Ardingly, West Sussex RH17 6TN, UK

**Keywords:** fruit crop, maternal environment, plant genetic resources, seed storage, sweet orange

## Abstract

Intra-specific variation in seed storage behaviour observed in several species has been related to different maternal environments. However, the particular environmental conditions and molecular processes involved in intra-specific variation of desiccation tolerance remain unclear. We chose *Citrus sinensis ‘bingtangcheng’* for the present study due to its known variability in desiccation tolerance amongst seed lots. Six seed lots of mature fruits were harvested across China and systematically compared for drying sensitivity. Annual sunshine hours and average temperature from December to May showed positive correlations with the level of seed survival of dehydration. Transcriptional analysis indicated significant variation in gene expression between relatively desiccation-tolerant (DT) and -sensitive (DS) seed lots after harvest. The major genes involved in late seed maturation, such as heat shock proteins, showed higher expression in the DT seed lot. Following the imposition of drying, 80% of stress-responsive genes in the DS seed lot changed to the stable levels seen in the DT seed lot prior to and post-desiccation. However, the changes in expression of stress-responsive genes in DS seeds did not improve their tolerance to desiccation. Thus, higher desiccation tolerance of *Citrus sinensis ‘bingtangcheng’* seeds is modulated by the maternal environment (e.g., higher annual sunshine hours and seasonal temperature) during seed development and involves stable expression levels of stress-responsive genes.

## 1. Introduction

Intra-specific variation in desiccation tolerance (DT) in non-orthodox seeds has been observed in several plant species. Previous studies have suggested that adaptation to different environmental conditions (i.e., the maternal environment) and genetic background possibly influence such variability in desiccation tolerance [1,2,3,4]. For example, two subspecies of *Quercus ithaburensis* produced seeds that exhibited different responses to drying [4] and two varieties of *Camellia sinensis* from Kunming (*Camellia sinensis* var. *sinensis*) were less desiccation sensitive than two other collections from Puer and Lincang (*Camellia sinenesis* var. *assamica*) even though all seed lots appeared to be at a similar well-developed stage of maturity. In *Camellia sinensis*, tolerance of dry seasons and genetic background may have contributed to the difference observed in seed DT [3]. In *Acer pseudoplatanus*, seeds from populations of southern Europe showed higher DT than those of northern populations [2]. Other trait parameters of the northern populations, such as lighter seed weight and higher embryo water content, suggested the differences were related to heat sum (determined by temperature and sunshine hours) during seed development [2]. These findings corroborated earlier studies on recalcitrant seeds of *Aesculus hippocastanum*, which showed heat sum-dependent developmental status predicted differential seed quality traits at the point of dispersal across Europe [5]. Thus, intra-specific variation in seed DT can be affected by environmental conditions during seed development.

Orthodox seed development can be divided into three phases: embryogenesis, seed filling and late seed maturation drying [6]. Such seeds generally gain their full DT during the last stage, while non-orthodox seeds tend not to reach this developmental stage and thus have varying levels of desiccation sensitivity (DS). During the late seed maturation stage, key biochemical changes occur such as photosynthetic pigments degradation, and the accumulation of heat shock proteins and soluble sugars, including sucrose, arabinose, galactose and raffinose family oligosaccharides (RFO) [7,8]. Consequently, the particular stage of seed maturity largely influences DT [6]. In the model species *Arabidopsis*, a major metabolic switch to the accumulation of distinct sugars, organic acids, nitrogen-rich amino acids and shikimate-derived metabolisms happens during the transition from the reserve accumulation to seed maturation-desiccation stages [7]. Particular gene families have also been implicated in the development of seed DT. For example, WRKY family genes are positive regulators involved in biotic defence. Defence process-related GO categories were found to be highly enriched during seed maturation [8]. How sensitive this metabolic shift and associated (gene expression) molecular processes are to particular maternal environmental conditions is still unclear. As for the germination trait, intra-specific variation in DT could also be significantly impacted by environmental conditions during development [9]. In this regard, *Citrus sinensis ‘bingtangcheng’* is an ideal study material to explore a modulating role of the maternal environment on seed DT as the species has been widely planted across China and genomic resources are available [10].

*Citrus sinensis* is a major cultivated species of the Rutaceae family [11]. *Citrus* is believed to have originated from the southeast foothills of the Himalayan region that includes the eastern area of Assam, northern Myanmar and western Yunnan [10,12]. The cultivation of this fruit crop started at least 4000 years ago [13,14], and *Citrus* species are a prime source of vitamin C for human nutrition. Conserving the genetic resources of the *Citrus* genus is important to plant breeding, due to the presence of desirable traits in its crop wild relatives. However, there is considerable inter-specific variation in seed storage behaviour (particularly DT). Of the twenty-six species listed on the Seed Information Database (Seed Information Database [SID], RBG, Kew, 2022), only three are listed as having orthodox seeds, while the rest have varying levels of DS. Even for the same species (e.g., *Citrus histrix* DC.), findings have indicated orthodox and recalcitrant storage behaviour in separate studies (SID, RBG, Kew, 2022). The seeds of sweet orange (*C. sinensis ‘bingtangcheng’*) are reputed to be oily and partially DT.

In this study, we investigated the differences in DT of *C. sinensis ‘bingtangcheng’* from six locations in China across an environmental gradient. Then we examined the transcriptome of seed lots from the two ends of the DT range to dissect the role of seed maturation and stress related genes in DT. Finally, we discussed whether simply switching on key genes is enough to confer DT.

## 2. Results

### 2.1. Desiccation Tolerance (DT) of Citrus sinensis ‘bingtangcheng’ Seed Lots from Different Locations

To gain a better comparison of DT among *C. sinensis ‘bingtangcheng’* seeds, six seed lots were collected from latitudes from 24°12′ N to 29°45′ N and altitudes varying from 164 to 1133 m a.s.l. (Table 1). The total germination (TG) percentage before desiccation was above 90% for all seed lots and it declined during desiccation (Figure 1). Seeds from Chongqing (Figure 1A) and Shaoguan (Figure 1B) showed the lowest and highest relative DT, compared with seed lots from Huaning, Huaihua, Guilin and Ganzhou (Figure 1C). After 6 d of drying to 2.7% MC, TG in Shaoguan seed lot was still 67% (Figure 1B). Seed lots from Huaning, Guilin and Ganzhou also showed relatively high DT, with 50%, 40% and 40% TG, respectively, when dried to <5% MC. However, seeds from Huaihua and Chongqing showed low relative desiccation tolerance, with 20% and 29% TG when dried down to 4.9% and 7.7% MC, respectively. A co-plot of TG (probit) against seed MC revealed that when germination was 5.5 probit (i.e., 69% TG at 2.7% MC for the Shaoguan seed lot), the interpolated MCs (i.e., a measure of DT) were 4.5%, 7.9%, 9.2%, 13.6%, 22.3% and 25.5% MC for Shaoguan, Guilin, Huaning, Ganzhou, Huaihua, and Chongqing seed lots, respectively (Figure 1C). These results confirm that Shaoguan was the seed lot with the highest DT and Chongqing the one with the lowest. These findings then framed the molecular studies.

### 2.2. Correlation Studies between DT of Citrus sinensis ‘bingtangcheng’ Seed Lots and Other Seed Traits

Seed traits like seed mass (mg), seed dry mass (mg) and initial MC (%) were tested as co-correlants of DT (indicated as the MC at which germination was interpolated to be probit 5.5: the lower this MC, the higher the DT). Variation in seed mass (mg) of the seed lots was normally distributed (Appendix A). Mean seed mass of *C. sinensis ‘bingtangcheng’* from Chongqing was the smallest (63.6 mg), while Ganzhou had the largest seed mass at 120 mg (*p* < 0.05) (Table 1). Seed dry mass did not correlate with DT (r = −0.4) (Figure 2K). The initial MCs of the seeds ranged from 41.8% for Ganzhou to 57% for Chongqing (Table 1), and this trait showed strong negative correlation with seed DT (r = 0.64) (Figure 2L). However, although seed lots from Huaning and Huaihua contained similar MCs of 48.7% and 49.4% respectively, their DT differed (Table 1).

### 2.3. Correlation Studies between Seed DT and Meteorological Data

The environmental conditions of the six locations from which *Citrus sinensis ‘bingtangcheng’* fruits were harvested were slightly different (Figure 3, Appendix A). Altitude did not correlate with seed DT (r = −0.20) (Figure 2J). The average temperature in Shaoguan was highest from December to May, while that in Chongqing was one of the lowest during this time (Figure 3B). Seed DT showed a strong positive correlation with the average temperature over this period (r = −0.72) (Figure 2B). The average precipitation was greatest in Guilin (157.3 mm), while Chongqing and Huaning received precipitation below 100 mm: 75.7 mm and 92.3 mm, respectively. Overall, the average annual rainfall (r = −0.39), the average rainfall from December to May (r = −0.34) and that from June to November (r = −0.28) showed only weak positive correlations with seed DT (Figure 2D–F). However, annual average sunshine hours did correlate positively with seed DT (r = −0.60; Figure 2G). Annual average sunshine hours from December to May and from June to November were also positively correlated with seed DT, with r = −0.39 (Figure 2H) and r = −0.53 (Figure 2I) respectively. Besides, analysis of temperature and sunshine hours during the last month before fruit harvesting (November) was analyzed and showed strong correlation (r = −0.59 and r = −0.80 respectively) (Appendix A). Sunshine hours and temperature are two important factors for developmental heat sum, which impacts seed development and DT [2]. Thus, our results showed that variation in sunshine hours and temperature between Chongqing and Shaoguan resulted in different levels of seed DT at the point of fruit maturity.

### 2.4. Transcriptome Analysis of DS and DT Seed Lots from Different Locations

Because of the known ecological correlates for varying DT between Chongqing (DS) and Shaoguan (DT), we then explored the molecular factors involved in this variation in stress tolerance. We compared gene expression during desiccation of these two seed lots (Figure 4). We used “FDR < 0.05 and |log_2_FC| ≥ 1” as the criteria for significant difference in gene expression (DEGs). The contrast between the two different seed lots before desiccation (DS0 vs. DT0) showed the largest number of DEGs across all comparisons, with 2590 DEGs up-regulated and 1009 DEGs down-regulated (Appendix A, Figure 4A,B). During the drying process of the DS seed lot (DS0 vs. DS6), 2779 DEGs were identified, of which 1145 were up-regulated and 1634 down-regulated (Appendix A, Figure 4A,B). In contrast, only 444 DEGs were identified when a relatively DT seed lot was compared before and after the desiccation process (DT0 vs. DT6), with 26 up-regulated genes and 418 down-regulated genes (Appendix A, Figure 4A,B). Principal component analysis (PCA) was used to reveal the overall variance among DEGs (Figure 4C). Principal component 1 (PC1) explained the separation of the different seed lots. These results suggest that large transcriptomic differences already existed between the two different seed lots before drying. In addition, the dehydration process triggered considerable gene expression change in the DS seed lot, while not much affecting the DT seed lot.

### 2.5. Gene Expression Patterns of DT Citrus sinensis ‘bingtangcheng’ Seeds Are Characteristic of a Late Seed Maturation Stage

With respect to the genes involved in DT in *Citrus sinensis ‘bingtangcheng’* seeds, we first analysed the expression of transcription factors (TFs), which play critical roles in regulating this trait. In the present study, 238 unigenes encoding TFs belonging to 43 families showed significant differences in expression when comparing two seed lots before desiccation (DS0 vs. DT0; Figure 5A, Appendix A). Focusing on DS seeds during dehydration (DS0 vs. DS6; Figure 5B, Appendix A), there were 161 unigenes encoding TFs for which gene expression was significantly changed. In contrast, only nine unigenes encoding TFs showed significant changes following the drying of the DT seed lot (i.e., DT0 vs. DT6; Figure 5C, Appendix A). The four TF families with the highest number of DEGs between DS0 vs. DT0 and DS0 vs. DS6 were ERF, NAC, MYB and WRKY (Figure 5). 

In order to analyse the role of specific late seed maturation genes, we compared our results to a previous report on late maturation of orthodox (fully DT) dicot seeds [6] and identified 82 unigenes involved in desiccation tolerance in *C. sinensis ‘bingtangcheng’* (Figure 6). These included genes encoding HSPs (41 unigenes), late embryogenesis abundant (LEA) proteins (3 unigenes), WRKYs (27 unigenes), as well as genes involved in the degradation of photosynthetic pigments (6 unigenes), or non-reducing sugar metabolism (5 unigenes). The expression levels of these unigenes in the two different seed lots (DS0 vs. DT0) are summarized in Appendix A. Among them were 55 unigenes with up-regulated and 27 unigenes with down-regulated expression in DT vs. DS seeds.

Regarding heat shock protein families, one unigene was predicted to encode for a heat shock factor 24-like protein (HSF24-like), seven for small heat shock proteins (sHSPs), three for HSP60 family proteins, seven for HSP70 family proteins, and one each for HSP90 and HSP100 family proteins. There were 30 HSP unigenes showing high expression levels in the DT seed lot (DT0), with 11 significantly up-regulated compared to the DS seed lot (DS0) (Figure 6B). Eleven HSP unigenes showed lower expression in the DT seed lot, but not significantly so (Figure 6B). Only three LEA-related unigenes were detected, with similar (high) expression in both seed lots (Figure 6B).

With respect to soluble sugars, four unigenes encoding galactinol synthase (two significantly) and raffinose synthase proteins showed higher expression in the DT seed lot (compared to DS) before drying (Figure 6B, Appendix A). The degradation of photosynthetic pigments is the most visible change associated with late seed maturation and six related genes were detected in the present study. Four unigenes encoding carotenoid cleavage dioxygenases showed higher transcript levels in the DT seed lot (DT0), with two significantly different from the DS seed lot (DS0) (Figure 6B, Appendix A). In addition, one gene encoding chlorophyll b reductase NYC1 showed significantly lower expression in the DT seed lot (Figure 6B).

In addition, our study identified 27 differentially expressed WRKY unigenes for the DS0 vs. DT0 contrast, among which four WRKY unigenes had significantly higher expression in DT seed lots. Overall, two thirds of WRKY unigenes showed higher expression in the DT seed lot, while one third showed lower expression (Figure 6B, Appendix A).

Together, the data above indicate that the seed lots from Shaoguan (DT0) and Chongqing (DS0) differed in developmental stage, with the seeds from Shaoguan showing gene expression characteristics associated with later seed maturation.

### 2.6. DT Citrus sinensis ‘bingtangcheng’ Seeds Show Stable Levels of Stress Responsive Genes during Drying

To further understand the types and function of genes involved in desiccation tolerance, DEGs of the two *Citrus sinensis ‘bingtangcheng’* seed lots, Shaoguan (DT) and Chongqing (DS), were analysed for enrichment of GO terms (Figure 7). For the biological process category, the top 10 significantly enriched GO terms of the DS seed lot during desiccation are listed in Figure 7C, with the top one being ‘response to stress’. There were 165 unigenes associated with this response-to-stress category, and the expression trends of those genes were compared in detail (Figure 7A,B, Appendix A). Of the 165 unigenes, 87 were significantly up-regulated and 78 down-regulated following drying in the DS seed lot (DS0 vs. DS6). In contrast, none showed significant changes following desiccation in the DT seed lot (DT0 vs. DT6). However, among those 165 in the response-to-stress category, the trend for 135 unigenes that changed in the DS seed lot was towards the level of the DT seeds. For example, for an HSP20-like chaperone superfamily protein (Unigene Cs8g19490), the value of FPKM increased in the DS seed from 852 to 2640 during the dehydration process, while the value in the DT seed lot was already at 1759 before drying (Appendix A). Another example is Cytochrome P450 (orange1.1t05138), for which the value of FPKM decreased from 72.23 to 3.36 in the dried DS seed, while the value in the DT seed lot was already at 3.71 before drying. These results show that DT *Citrus sinensis ‘bingtangcheng’* seeds have stable levels of stress-responsive genes. Moreover, the expression of 80% of stress-responsive genes in the DS seed lot during drying changed to the levels seen in the DT seed lot prior to dehydration.

## 3. Discussion

### 3.1. Desiccation Tolerance of Seeds from Mature Citrus sinensis ‘bingtangcheng’ Fruits and Relationships with Other Seed Traits and Environmental Conditions

It is extremely challenging to judge seed maturity in fleshy fruits. Thus, the relationship between seed and fruit maturation in fleshy fruits has rarely been studied, some exceptions being tomato and cucumber [15]. In contrast, the metabolic, hormonal and biochemical changes occurring during fruit development have been extensively studied [16]. For example, fruit tissue development seems regulated by SBP-box genes and their putative MADS-box promoter targets. If the gene at the ripening inhibitor (rin) locus is mutated, the fruit remains firm and unripe for extended periods [17]. A recent study on *Citrus* reported that seeds can show high viability before fruit maturation is reached [18]. However, whether the acquisition of high germination in seeds correlated with DT was not explored. In our case, although *Citrus sinensis ‘bingtangcheng’* fruits achieved maturation as defined within a commercial production setting and total germination before desiccation was high for all seed lots (>90%), seeds exhibited a large variation in DT, indicating potential differences in seed maturation between fruits of apparently similar maturity (Figure 1, Table 1).

Seed mass (or embryo mass) has been used as a surrogate measure of seed maturation and seed DT, with larger seeds indicating greater maturity [2,3]. For example, *Acer pseudoplatanus* fruits/seeds harvested across Europe with greater developmental heat sum, i.e., from southern rather than northern locations, were heavier and more mature at the point of natural dispersal [2]. Although the average dry seed mass of the most DS seed lot (Chongqing) and DT seed lot (Shaoguan) were 63.6 mg and 100.2 mg, respectively (Table 1), we found no overall correlation between dry seed mass and DT in the present study (Figure 2). A dry mass of 137 mg for *C. sinensis ‘bingtangcheng’* seeds reported by Hong et al. (2001) was higher than all seed lots analyzed in our study, and drying to 4% MC decreased their viability to 10% [19]. In contrast, the Shaoguan (DT) seed lot in our study retained 67% germination when dried to c. 3% MC (Figure 1), thereby showing higher DT irrespective of the seed mass. Similarly, seed initial MC has also been used as a surrogate measure of seed DT, with lower MCs indicating greater maturation [3]. For example, *Camellia sinensis* seeds with lower initial MC collected from three locations in China were more DT [2]. Although initial MC showed strong negative correlation with DT (Figure 2L), prediction of the precise level of DT is not certain. For example, Huaihua seeds (initial MC 49.4%) had 20% germination when dried to 5% MC while seeds from Huaning with the same initial MC (48.7%) had 50% germination after drying to <5% MC. However, the environmental conditions in those two places showed great differences (Appendix A), suggesting that a number of factors are at play during the development of DT, not just the seed harvest MC.

We then considered how much the environmental conditions at the six locations from which ‘mature’ *C. sinensis ‘bingtangcheng’* fruits were harvested for this study might co-correlate with seed DT [20]. We found that average annual sunshine hours and temperature from December to May showed strong correlations with seed DT (Figure 2). *C. sinensis ‘bingtangcheng’* trees from Chongqing (DS) received the least sunshine hours throughout the year and experienced one of the coolest temperatures from December to May compared with the other locations (Figure 3). Such limiting environmental conditions may have resulted in the least seed development despite the fruits being ready to eat. As the development and ripening of fruit may share similar genes with the development of floral organs [21], it is possible that temperatures from December to May might have (negatively) affected flowering time, and therefore the length of post-flowering development achieved at the *C. sinensis ‘bingtangcheng’* population sites. In addition, it appears that the temperature and sunshine hours during the month before fruit harvest (November) are important for seed maturation (Appendix A). Future studies could address whether or not seed lots identified here as DS are indeed developmentally immature, or genetically less DT.

Precipitation level and timing has also been a co-correlant of seed DT, in coffee [1], tea [3] and African trees [20]. Less precipitation after seed dispersal tends to be associated with more DT. However, the present study on intra-specific differences in *C. sinensis ‘bingtangcheng’* seeds prior to dispersal did not find a relationship between precipitation and relative DT spanning a broad range of moisture levels.

### 3.2. Transcriptome Analysis before and after Drying of Two C. sinensis ‘bingtangcheng’ Seed Lots Differing in DT

Although *C. sinensis ‘bingtangcheng’* is considered to have partial DT, we showed relatively high DT in several of the six seed lots studied (Figure 1, Table 1). Transcriptional analysis revealed that considerable differences in gene expression already existed between the most DS (Chongqing) and DT (Shaoguan) seed lots following harvest, including a large number of transcription factors from ERF, NAC and MYB families, which are implicated in late seed maturation of orthodox seeds [22]. When comparing our results to the late maturation stage in *Arabidopsis* seeds, the top five TF families also included MYB, NAC and ERF [22], which indicates that these TFs are putative transcriptional regulators of DT in seeds. In particular, these TF families are documented to play vital roles in the regulation of ABA mediated and independent drought-signalling pathways [23]. Other important late seed maturation genes in orthodox seeds include those related to degradation of photosynthetic pigments and heat shock protein family genes (*HSPs*) [6]. For example, the presence of chlorophyll molecules in dry non-chlorophyllous seeds is considered a characteristic of immaturity and photosynthetic pigment degradation is a visible change during seed development [24,25,26]. Moreover, active chlorophyll has recently been identified as a key contributor to seed aging because the molecule is prone to oxidative stress in the absence of metabolic water [27]. In this study, *CCD1* and *CCD8* genes showed higher transcript levels in the DT compared to the DS seed lot prior to drying (Figure 6B). Carotenoid cleavage dioxygenases (CCD) regulate degradation of carotenoids and four CCDs (CCD1, CCD4, CCD7 and CCD8) are active during seed development in *Arabidopsis*. The *CCD1* gene is induced during the onset of the desiccation phase [28], and an increase in *CCD8* transcript levels resulted in a decrease in carotenoid contents in seeds [29].

The function of HSPs and sHSPs as molecular chaperons is to protect proteins from denaturation damage under both stressed and non-stressed conditions [6,30,31]. In this regard, we found that 11 genes encoding HSPs and sHSPs (e.g., *HSP70*, *HSP22.7*, *HSP18.2*, *HSP17.4*) showed significantly higher expression in the DT seeds compared to the DS seeds prior to desiccation (Figure 6B, Appendix A). HSP families are suggested to have a protective function in seed longevity, and the expression of *HSPs* and *sHSPs* is up-regulated during late seed maturation [32,33]. Longer-term survival in cold, dry storage is known for seeds of some *Citrus* species [34], and the expression of HSP might suggest intra-specific variation in *C. sinensis ‘bingtangcheng’* seed lot storage in the dry state. But this remains to be explored.

Improvement of desiccation resistance with high level of HSPs is a universal phenomenon, existing in microorganisms, and cells of plants and animals. For example, the resistance to desiccation of *Azotobacter vinelandii* largely depends on HSP20 which might prevent the aggregation of proteins caused by the lack of water [28]. Moreover, higher constitutive levels of HSP74 in the foot tissue of land snail were correlated with higher desiccation resistance [35]. We discovered that the transcript levels of many HSP genes (including *HSP70*, *HSP26.5*, *HSP20*, *HSP18.2*, *HSP17.4*) were significantly increased during the desiccation process in the DS seed lot. In contrast, in the DT seed lot HSP gene transcript levels were already high and did not change in response to drying (Appendix A). Another group of genes that significantly increased in the DS seed lot during the desiccation process was peroxidase genes (e.g., fold change > 9, Appendix A). Peroxidase, such as ascorbate peroxidase, is considered one of the central ROS detoxification enzymes that function during the development of thermotolerance in plants [36]. Enrichment in stress-responsive genes, such as *HSP* and oxidative stress-related genes, was also observed during acquisition of desiccation tolerance in *Medicago truncatula* during seed development [37]. However, whilst drying activated changes in expression of stress-responsive genes in DS seeds, there was no concomitant improvement in their DT, as assessed by germination tests (Figure 1, Table 1). This suggests that the transient increase in transcripts of potential importance for DT is insufficient to ensure DT in DS seeds. One potential explanation could be that the imposed drying regime was too extreme (RH too low and time period too short), stimulating the high expression of genes that respond to stress, without this translating into sufficient protein synthesis. In this context, different drying regimes or desiccation tolerance re-induction methods, e.g., under higher RH or water potentials [38,39] could be tested in future studies. Now that the citrus genome has been annotated [9,11] and new plant breeding techniques, such as genome editing, are available, many further mechanistic studies can be pursued. For example, Ruby (a MYB-like TF) and its promoter 3′LTR which are one of the main genes responsible for the control of purple pigmentation in citrus fruits [40] could be used to compare fruit maturation across the six sites used in this study. Furthermore, the CRISPR/Cas12a system has already been examined for editing in grapefruit [41]. Future studies on seed DT should embrace these, and other, new plant breeding techniques.

## 4. Materials and Methods

### 4.1. Seed Material

Fruits of *Citrus sinensis ‘bingtangcheng’* from Huaning, Huaihua, Shaoguan, Guilin, Ganzhou and Chongqing were collected at the point of natural maturity (yellowing and softening) during the winter of 2014 (end November–end December) and used within 6 weeks of harvest. 

Fruits of *Citrus sinensis ‘bingtangcheng’* were collected from approximately 50 trees at each location and air freighted to the Germplasm Bank of Wild Species of China in Kunming. On arrival, fruits were kept in a cool place (about 10 °C, 8/16 h light/dark) for a maximum of 3 days before all the seeds were removed from the fruit. Cleaned seeds were stored in plastic bags at 5 °C for no more than 4 weeks as the experiments were started. The bags were opened at least once a week to allow for exchange of air. Only seeds with a healthy appearance (i.e., no apparent fungal or bacterial infection and mechanical damage) were pooled and randomly selected for the experiments.

### 4.2. Desiccation Treatments and Germination Test

Seeds were dried over silica gel (5:1 silica gel to seed weight) in sealed plastic bags at 15 °C for up to 6 days. The silica gel was replenished every 24 h and the bags ventilated. At each sampling time (desiccated for 0, 3, 4, 5, 6 d), four replicates of 10 seeds were withdrawn, three of which were used for the germination test. The remaining 10 seeds were used for moisture content (MC) determination. Seed mass values for the seed lots were calculated from MC determinations (*n* = 50 seeds).

For the germination test, each replicate of 10 seeds was placed in a 90 mm petri dish containing 1% water agar. Seed coats were removed before germination. The test was run at 30 °C under a day/night cycle of 8/16 h. Germination was defined as radicle emergence of at least 2 mm and it was scored regularly. Total germination (TG) was monitored for 35 d from the onset of imbibition. As no further germination occurring during the final 7 d period, TGs at this time were therefore considered final. Cut-tests were performed on the remaining non-germinated seeds and the vast majority was found to be soft and assessed as being inviable. Seed germination was expressed as a percentage of the total number of seeds sown. The MCs (10 randomly selected seeds/test) were determined gravimetrically by drying the seeds at 103 °C for 17 h (ISTA 2022) (https://www.seedtest.org (accessed on 6 Jun 2022)).

### 4.3. Environmental Conditions at the Sampling Sites

Meteorological data (Appendix A) were obtained from the National Meteorological Information Centre (data.cma.cn (accessed on 12 May 2019)) which provides monthly average temperature, precipitation data and sunshine duration hours recorded from 1981 to 2010 for each location. 

### 4.4. Isolation of RNA, Library Construction and Sequencing

Paired-end transcriptome sequencing (RNA-Seq) was employed to analyse gene expression in seeds from Chongqing (DS) and Shaoguan (DT) at two desiccation stages: dried for 0 d (DS0 and DT0) and for 6 d (DS6 and DT6). Total RNA was isolated using the Omega Plant RNA Kit (Omega Bio-Tek, Norcross, GA, USA) according to the manufacturer’s protocol. Three total RNA samples were isolated for each desiccation stage (0 and 6 d) and each sample was isolated from at least 10 ground seeds. The RNA quality was verified by an Agilent 2100 Bioanalyzer (Agilent Technologies, Palo Alto, CA, USA) and checked using RNase free agarose gel electrophoresis. One sample from each time point with the best quality of RNA was selected for RNA-Seq. High quality RNA was enriched by Sera-Mag oligo (dT) beads (Thermo Scientific, Indianapolis, IN, USA). The enriched mRNA was fragmented into small pieces by a fragmentation buffer and reverse-transcribed to cDNA using NEBNext Ultra RNA Library Prep Kit for Illumina (NEB 7530, New EnglandBiolabs, Ipswich, MA, USA). The cDNA was end-repaired with an adapter primer attached and subsequently adaptor-ligated by the addition of a specific adapter, following the manufacturer’s protocol (Illumina Inc. San Diego, CA, USA). The cDNA fragments were sequenced using Illumina HiSeq2500 by Gene Denovo Biotechnology Co. (Guangzhou, China).

### 4.5. Gene Expression and GO Enrichment Analyses

Expression abundance was quantified using ‘Fragment Per Kilobase of transcript per Million mapped reads’ (FPKM) (Cufflinks software, version 2.2.1) [42]. Differential expression analysis was performed by DESeq2 software (version 1.4.5) for three different contrasts (DS0-vs-DT0, DS0-vs-DS6 and DT0-vs-DT6) [43]. The edgeR software (version 3.6.8) was used to test significant differences in the expression of count data as described previously [44,45] and the selected dispersion value was 0.01. When compared with a reference sequence (CsiDB2011_11), 74.2% of known genes were detected in the four samples, and the number of DEGs between different samples were analysed (Appendix A). Differentially expressed genes/transcripts (DEGs) were counted as the genes/transcripts with a false discovery rate (FDR) below 0.05 and absolute log_2_ fold change ≥ 1. All DEGs were mapped to Gene Ontology (GO) terms through the Gene Ontology database (https://www.geneontology.org/ (accessed on 6 May 2016)) for GO enrichment and pathway analysis. Gene numbers were calculated for every term, with significantly enriched GO terms in the DEG list (compared to the genome) defined by hypergeometric test. The calculated *p*-values were subjected to FDR correction, utilizing FDR ≤ 0.05 as a threshold.

### 4.6. Data Analysis and Presentation

Seed dry mass and initial moisture content were statistically analysed using IBM SPSS Statistics 25.0 software, version SPSS 25.0. Principal component analysis was also conducted with SPSS 25.0 software, version Origin 7.0 (SPSS, Chicago, IL, USA). The data conforming to normal distribution and homogeneity of variance was tested using one-way ANOVA followed by pairwise comparisons of LSD to identify difference. In the results, significant differences (*p* < 0.05) are indicated by different letters. The Origin 7.0 software (Origin 7.0, OriginLab, Northampton, MA, USA) was used to draw figures and for linear fitting (e.g., to analyse correlation in Figure 2. The germination percentage values were converted to probit germination (Figure 1C) use online sources, e.g., http://data.kew.org/sid/viability/convert.jsp (accessed on 3 June 2021).

## 5. Conclusions

While differences in DT have been reported for the same species from a wide range of locations [3,5] little is known still about the translation of maternal environmental conditions through molecular processes into the acquisition of physiological DT. To gain insight, we systematically compared DT of seed lots harvested from mature *Citrus sinensis ‘bingtangcheng’* fruits from six locations across China (Figure 1 and Figure 2, Table 1). We found that these ‘mature’ fruits yielded seed lots that all had acquired germination competence but varied widely in their DT. The annual average sunshine hours and average temperature from December to May showed strong correlations with seed DT (Table 1, Figure 2). In the future, we could use of real-time data loggers placed at each site to detect higher co-correlants environmental factors of seed DT.

Transcriptional analysis revealed that large differences in gene expression already existed between the most DS and DT seed lots following harvest. Differentially expressed genes implicated in late seed maturation of orthodox seeds showed a large number of transcription factors (e.g., ERF, NAC, MYB and WRKY families), and a higher expression of carotenoid cleavage dioxygenases and HSPs in the more tolerant seed lot. Following the imposition of drying, about 80% of stress-responsive genes in the DS seed lot changed to the stable levels seen in the DT seed lot prior to and post-desiccation (Figure 7). However, the changes in expression of stress-responsive genes in DS seeds did not improve their DT. Future studies could explore whether DS seeds harvested from ‘mature’ fruits are developmentally immature, and/or genetically less DT.

## Figures and Tables

**Figure 1 ijms-24-07393-f001:**
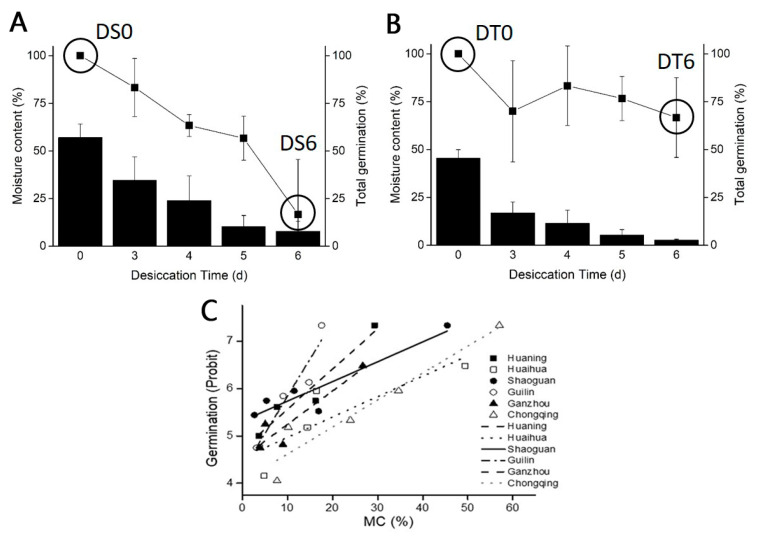
Effect of drying on seed parameters of *Citrus sinensis ‘bingtangcheng’* seed lots. Total germination (TG, filled squares) response of (**A**) Chongqing (desiccation sensitive, DS) and (**B**) Shaoguan (desiccation tolerant, DT) seed lots (*n* = 3 biological replicates of 10 seeds each) after drying for up to 6 days (DS6, DT6) to different moisture contents (filled bars). Whole seed moisture contents (MC) were determined for three biological replicates of 10 seeds each. Data represent means ± SD. (**C**) Plot of the relationship between MC and TG (probit scale) for Huaning (■), Huaihua (□), Shaoguan (●), Guilin (○), Ganzhou (▲), Chongqing (△). Data represent mean TG at each MC.

**Figure 2 ijms-24-07393-f002:**
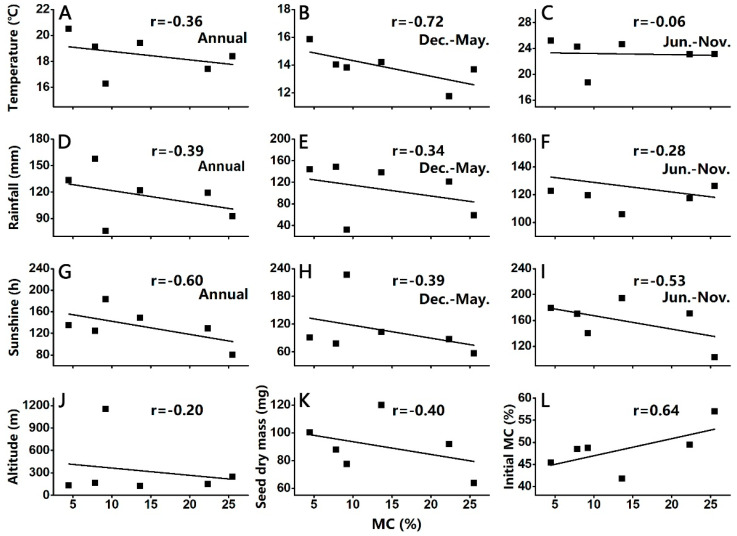
Correlation between the seed MC at which germination was interpolated to be probit 5.5 and *Citrus sinensis ‘bingtangcheng’* seed lot characteristics as well as environmental conditions of the collecting sites. (**A**–**C**) Average temperature (°C), (**D**–**F**) average rainfall (mm), (**G**–**I**) sunshine hours (**H**), (**J**) altitude, (**K**) seed dry mass and (**L**) initial moisture content. Climate conditions were averaged for Dec.–May (after fruit dispersal and before flowering) and Jun.–Nov. (after flowering).

**Figure 3 ijms-24-07393-f003:**
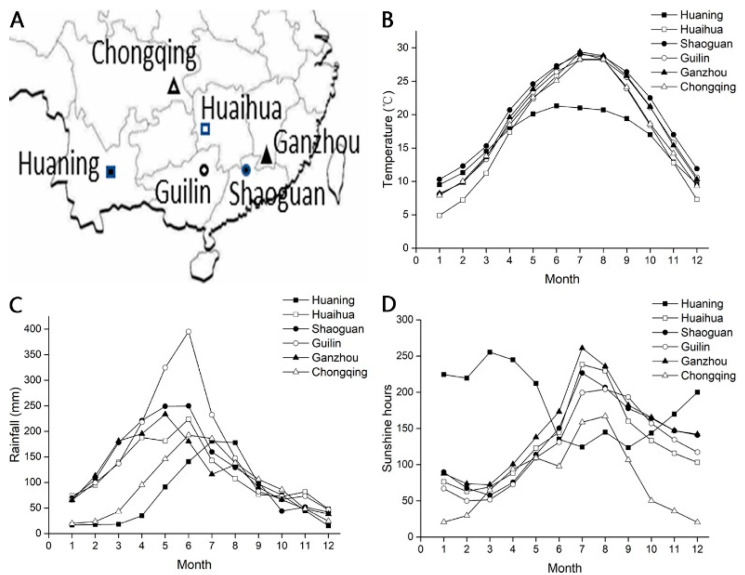
*Citrus sinensis ‘bingtangcheng’* collection site environment details. (**A**) map of locations of the six collection sitesacross China, (**B**) Average temperature (°C), (**C**) Average rainfall (mm) and (**D**) Average sunshine duration hours recorded from 1981 to 2010 for Huaning (■), Huaihua (□), Shaoguan (●), Guilin (○), Ganzhou (▲) and Chongqing (△).

**Figure 4 ijms-24-07393-f004:**
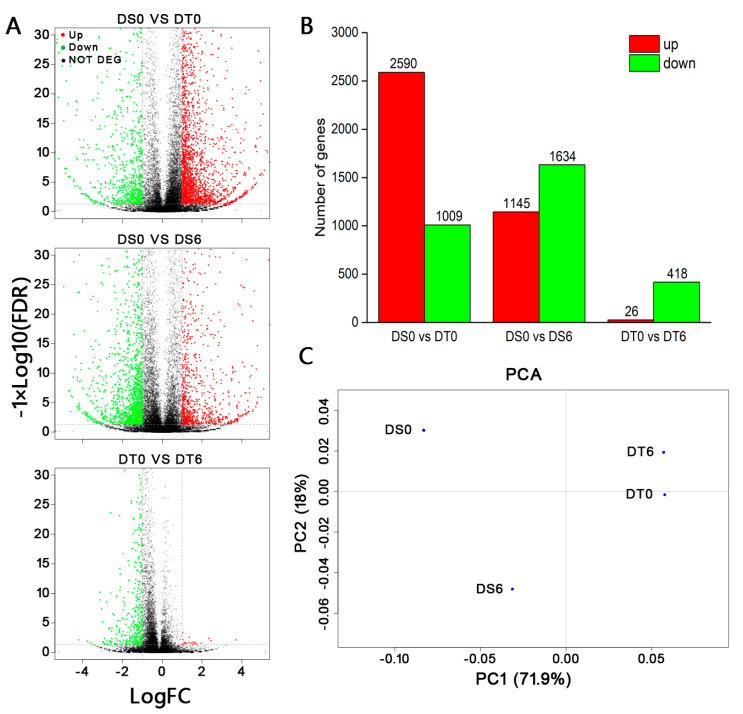
Differentially expressed genes (DEGs: FDR < 0.05 and absolute log_2_ fold change ≥ 1) were identified for three different comparisons. (**A**) Volcano plots depicting DEGs between two seed lots before desiccation (DS0 vs. DT0) and within seed lots exposed to desiccation (DS0 vs. DS6, DT0 vs. DT6). Red dots indicate significantly up-regulated genes, green dots indicate significantly down-regulated genes, and black dots represent non-DEGs. (**B**) Up-(red) and down-regulated (green) DEGs for each comparison. (**C**) Principal component analysis for the four molecular analysis: DS0, desiccation sensitive seeds dried for 0 d, DS6, desiccation sensitive seeds dehydrated for 6 d, DT0, desiccation tolerant seeds dried for 0 d and DT6, desiccation tolerant seeds dehydrated for 6 d.

**Figure 5 ijms-24-07393-f005:**
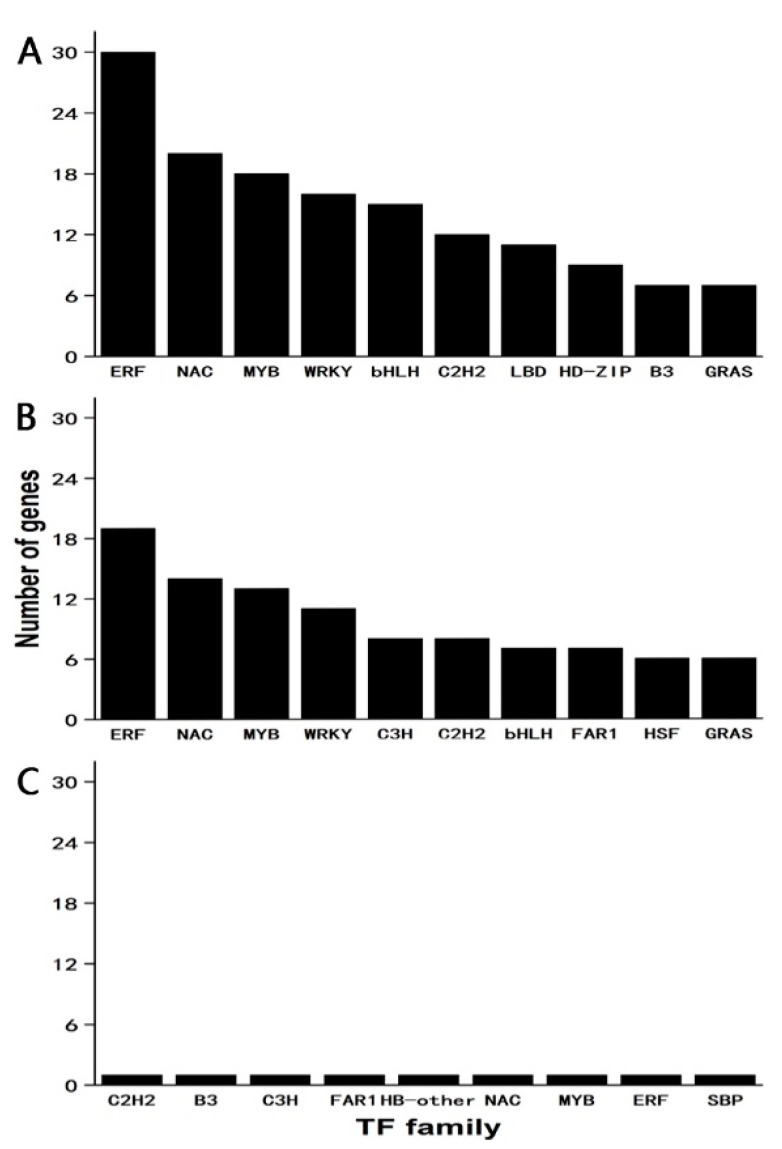
The number distribution of DEGs (FDR < 0.05 and absolute log_2_ fold change ≥ 1) that belong to the top 10 transcription factor families identified from three comparisons. (**A**) DS0 vs. DT0; (**B**) DS0 vs. DS6; (**C**) DT0 vs. DT6. DS0 signifies desiccation-sensitive seeds dried for 0 d; DS6: desiccation-sensitive seeds dehydrated for 6 d; DT0, desiccation-tolerant seeds dried for 0 d; DT6, desiccation-tolerant seeds dehydrated for 6 d.

**Figure 6 ijms-24-07393-f006:**
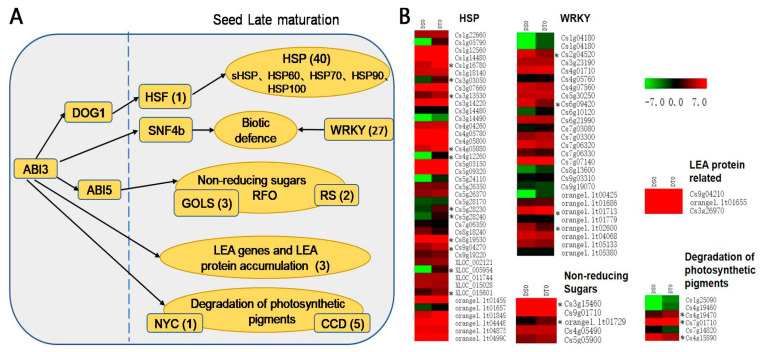
Late seed maturation related unigenes and their expression in *Citrus sinensis ‘bingtangcheng’.* (**A**) Putative unigenes involved in late seed maturation. The value in brackets indicates the number of unigenes annotated. (**B**) The heatmap shows the transcript level of each late seed maturation related unigene in the desiccation sensitive (DS0) and the desiccation tolerant (DT0) seed lots. The asterisks indicate the differentially expressed unigenes (FDR < 0.05 and absolute log_2_ fold change ≥ 1) between the two seed lots before desiccation (DS0 vs. DT0).

**Figure 7 ijms-24-07393-f007:**
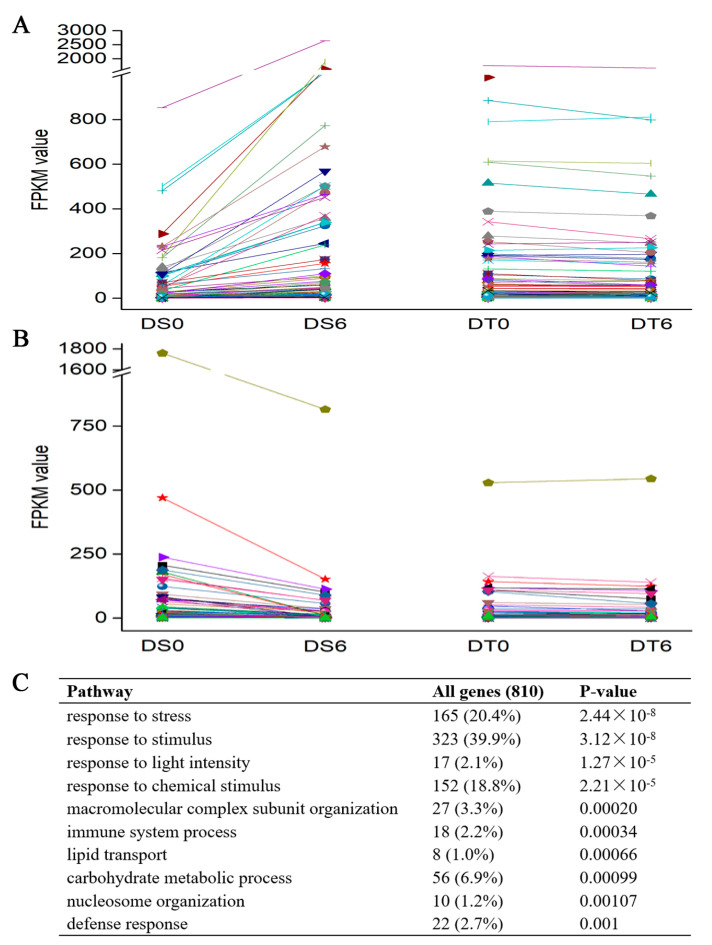
Analysis of stress-responsive genes that were differentially expressed (DEGs: FDR < 0.05 and absolute log^2^ fold change ≥ 1) in DS seeds following drying. (**A**) Expression levels of stress-responsive genes that were significantly up-regulated, and (**B**) significantly down-regulated in DS seeds following drying. (**C**) The top 10 enriched GO categories of DEGs (DS0 vs. DS6). DS0 signifies desiccation-sensitive seeds dried for 0 d; DS6, desiccation sensitive seeds dehydrated for 6 d; DT0, desiccation tolerant seeds dried for 0 d; and DT6, desiccation tolerant seeds dehydrated for 6d. The color lines represents the trend of gene changes from 0 d to 6 d.

**Table 1 ijms-24-07393-t001:** Details of the six seed lots of *Citrus sinensis ‘bingtangcheng’* studied. MC (%) at TG probit 5.5 is a measure of seed DT (the lower this MC, the higher the DT).

Seed Lot	Location	Altitude (m)	Collection Date	MC (%) at TG Probit 5.5	Whole Seed Dry Mass (mg)	Initial Moisture Content (%)
Huaning	24°12′ N; 103°07′ E	1133	5 December 2014	9.2	77.5 ± 2.5 ^d^	48.7 ± 1.4 ^b^
Huaihua	27°47′ N; 109°48′ E	244	9 December 2014	22.3	91.7 ± 3.5 ^b,c^	49.4 ± 2.6 ^b^
Shaoguan	24°41′ N; 113°49′ E	373	26 December 2014	4.5	100.2 ± 2.8 ^b^	45.4 ± 1.4 ^b,c^
Guilin	25°24′ N; 110°19′ E	164	31 December 2014	7.9	87.7 ± 3.1 ^c^	48.4 ± 2.1 ^b^
Ganzhou	25°70′ N; 115°20′ E	319	15 December 2014	13.6	120 ± 3.5 ^a^	41.8 ± 1.4 ^c^
Chongqing	29°45′ N; 106°22′ E	245	28 December 2014	25.5	63.6 ± 2.6 ^e^	57.0 ± 1.4 ^a^

Values are mean ± SE for 50 individual seeds. Values in the same column with a different letter are significantly different (*p* < 0.05).

## Data Availability

The authors confirm that the data supporting the findings of this study are available within the article and its Appendix A.

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
