# Peer review of "Intra-Specific Variation in Desiccation Tolerance of Citrus sinensis ‘bingtangcheng’ (L.) Seeds under Different Environmental Conditions in China"

_ijms, 2023, doi:10.3390/ijms24087393_

Round 1

Reviewer 1 Report

This submission presents very interesting and valuable findings regarding  potential mechanisms causing intra-specific variation in desiccation tolerance of Citrus seeds. Difference in desiccation tolerance among seed lots of the same species grown in different geographical locations and under different environmental conditions is a known but yet not well understood phenomenon. In addition, it is quite difficult to study. The authors performed a comprehensive analysis of the desiccation tolerance of Citrus sinensis seeds collected from six provinces in China and explored its linkage to seed traits and environmental conditions. Furthermore, transcriptomic analysis was performed for most desiccation sensitive and tolerant seed lots  before and after desiccation. The study is well-though, with a perfectly logical structure and clear description of both methodology and the results. These findings will make a great contribution to our overall understanding of internal and external factors determining the development of desiccation tolerance in non-orthodox seeds. I suggest acceptance after minor revision to fix small typos and errors in the presentation.

1.     Please move Table 1 after its first mention in the text.

2.     Line 97. Could you explain for the readers why you choose 5.5 probit (69% germination) as a threshold, and not lower or higher values?

3.     Figure 1 legend. Last sentence. “at each MC”?

4.     Lines 189-191. Sentence starting with  “Focusing on DS..” is incomplete?

5.     Figure 6. Bottom right gene cluster. The name should be “DegrAdation of photosynthetic pigments”? something is missing here

6.     In Discussion, 3.1., you don’t say much about correlation between seed initial MC and their desiccation tolerance, but discuss other traits and environmental factors with lower or no correlation. Please add initial MC to the discussion.

7.     Materials and methods. Line 390. “cool place” – please add temperature and light/dark conditions.

8.     Line 467. “still know little about”.. “still little is known about”?

9.     Supplemental Tables S1-3 are not mentioned in the text or I have missed it. Please check.

10.  Supplemental data files 2,3, 5 and 6 have “DT1” or “DS1” in their names. Should those be “DT6” and “DS6” for the direct match to the results?

11.  A consideration: the correlation between the initial MC and seed desiccation tolerance remains uncertain to me. For example, seeds from Huaihua province (initial MC 49.4%) show 20% germination when dried to 5% MC. Seeds from Huaning province with the same initial MC (48.7%) shows 50% germination after drying to <5% MC. So, it depends on how you access desiccation tolerance.  I agree with the indication of DT that you follow in the paper, but it looks like initial MC is a dependent value which might be influenced by the environmental conditions. This is why I would like to see some discussion related to the initial MC in the Discussion section.

12.  For future research, it would be interesting to compare transcriptomic profiles of two desiccation-tolerant and two desiccation-sensitive seed lots from different provinces. Just to confirm that the DEGs you have found are not the result of some other unknown factors that varied depending on each province.

Author Response

Point1: Please move Table 1 after its first mention in the text.

Response 1: It has been moved after the first mention. Please see line 134.

Point 2: Line 97. Could you explain for the readers why you choose 5.5 probit (69% germination) as a threshold, and not lower or higher values?

Response 2: It has been added, please see line 106.

Point 3. Figure 1 legend. Last sentence. “at each MC”?

Response 3: Thanks for the correction. Please see line 121.

Point 4. Lines 189-191. Sentence starting with “Focusing on DS..” is incomplete?

Response 4: Thanks for correction. Please see line 210.

Point 5. Figure 6. Bottom right gene cluster. The name should be “DegrAdation of photosynthetic pigments”? something is missing here.

Response 5:Thanks for correction. Please see line 268.

Point 6. In Discussion, 3.1., you don’t say much about correlation between seed initial MC and their desiccation tolerance, but discuss other traits and environmental factors with lower or no correlation. Please add initial MC to the discussion.

Response 6: Thank you for the reviewer’s good suggestion. The discussion has been added. Please see line 334.

Point 7: Materials and methods. Line 390. “cool place” – please add temperature and light/dark conditions.

Response 7: Thanks for suggestion, the detail of cool place has been added, please see line 438.

Point 8: Line 467. “still know little about”.. “still little is known about”?

Response 8: Thanks for correction, please see line 511.

Point 9: Supplemental Tables S1-3 are not mentioned in the text or I have missed it. Please check.

Response 9: Thanks for reminding, it has been added into text, please see line 148, line 462.

Point 10.  Supplemental data files 2,3, 5 and 6 have “DT1” or “DS1” in their names. Should those be “DT6” and “DS6” for the direct match to the results?

Response 10: We really appreciated for the reviewer’s correction. DT1 should be changed to DT6 and DS 1 should be changed to DS6, in accordance with the main text. It has been changed in the supplemental data file.

Point 11: A consideration: the correlation between the initial MC and seed desiccation tolerance remains uncertain to me. For example, seeds from Huaihua province (initial MC 49.4%) show 20% germination when dried to 5% MC. Seeds from Huaning province with the same initial MC (48.7%) shows 50% germination after drying to <5% MC. So, it depends on how you access desiccation tolerance.  I agree with the indication of DT that you follow in the paper, but it looks like initial MC is a dependent value which might be influenced by the environmental conditions. This is why I would like to see some discussion related to the initial MC in the Discussion section.

Response 11: We really appreciated the reviewer’s contribution for the discussion part. Please see line 334.

Point 12:For future research, it would be interesting to compare transcriptomic profiles of two desiccation-tolerant and two desiccation-sensitive seed lots from different provinces. Just to confirm that the DEGs you have found are not the result of some other unknown factors that varied depending on each province.

Response 12: Thanks for the suggestion from the reviewer’s suggestion, we will plan to do more related experiments in the future.

Reviewer 2 Report

The article is well written and produces interesting findings, but before going any deeper into details, there is one key question, that has to be answered, that I have not found in the manuscript - are all of the seeds from the same lineage or variety? Just because all the seeds are from the same species doesnt guarantee (and in my experience it doesnt) that the main factors, that are studied have the main explainatory value. There might be large influence of genetic background and if there is no guarantee that all the plants are comming from same variety/lineage, no matter how statistically significant effects of other factors are, they do lack significance for general conclusions.

So authors should (as a first step before going any deeper into the review process) either specify if all the plants are from the same variety/lineage and if not, then provide specifications of all the different varieties/lineages and discuss known phenotipical differences from different varieties and their potential influence on results.

Author Response

Point 1: The article is well written and produces interesting findings, but before going any deeper into details, there is one key question, that has to be answered, that I have not found in the manuscript - are all of the seeds from the same lineage or variety? Just because all the seeds are from the same species doesnt guarantee (and in my experience it doesnt) that the main factors, that are studied have the main explainatory value. There might be large influence of genetic background and if there is no guarantee that all the plants are comming from same variety/lineage, no matter how statistically significant effects of other factors are, they do lack significance for general conclusions.

So authors should (as a first step before going any deeper into the review process) either specify if all the plants are from the same variety/lineage and if not, then provide specifications of all the different varieties/lineages and discuss known phenotipical differences from different varieties and their potential influence on results.

Response 1: We thank referee for his comments on this very important point. We confirm that all the seeds in the present study came from same variety called Citrus sinensis ‘bingtangcheng’. In order to make it clear, we already changed all the name in the paper.

Reviewer 3 Report

The manuscript of Chen et al. studies desiccation tolerance of different populations of Citrus sinensis in dependence on the growing conditions, i.e., environmental impact. Additionally, it discusses the role of stress-responsive gene expression in tolerance to desiccation. This study is well conducted and interesting, and the manuscript is composed in a competent and well organized style. I highly recommend this manuscript for publication. However, there are some minor complaints that should be addressed prior to publication. Please, refer to the pdf attached for more specific comments.

General concept comments

The introduction is well organized and concise, yet provides sufficient background for the research conducted. It also includes all relevant references. However, it would be good to add a short section considering desiccation tolerance related genes that are analyzed in this study and their function/role. This way you will avoid unnecessary explanations in Results section (see Lines 235-236, for example).

Results are well described and have continuous flow, which is very easy to follow and understand. All Figures and Tables are well described and properly referred to.

Discussion is well elaborated and connected with obtained results. All statements are supported by suitable references.        

Material and Methods are clearly written and well organized. All analyzes are described in sufficient detail.

Conclusions are concise and follow the results and discussion.

Specific comments

Please, check the pdf file attached.

Author Response

We really appreciated for the suggestions reviewer has been given, please see the new version that has been changed according to the reviewer's points.

Round 2

Reviewer 2 Report

Authors have answered my biggest concern, but there are still some issues to fix:

Figure 2:

There are really low correlation coeficients, that are quite often caused by outlier values. I would suggest some more complex statistical analysis than simple linear regression, either to eliminate the outlier values (like isolation forrest/random forrest) or regresion of higher order. Also, this should be discussed more in the text and underline the main explainatory factors and their possible physiological explanations.

In analysis regarding to meteorological parameters - authors should discuss if such long term view is relevant, or if (for example) effect of last season(s) would show other results. There might be effect of one extreme season (or even extreme month) that needs to be either taken into account or it has to be proven, that there were no such extremes with potential high impact on results (we had simmilar issues with our opium poppy drought stress experiments, even though poppy is one year plant, still there are known seasonal differences even in fruit trees).

Lines 323 - 333 and lines 334-343 are the same, check what should stay.

Line 522 WRKY related genes are not mentioned? Why?

Photographies ilustrating differencies in seeds would be useful (even as SM).

Line 475 - should be reformulated - its confusing about total RNA and polyA (mRNA), maybe mRNA enhancement or something like that

line 478 - reverse transcriptase kit (or enzyme) manufacturer should be stated

Author Response

Point 1: There are really low correlation coeficients (Fig.2), that are quite often caused by outlier values. I would suggest some more complex statistical analysis than simple linear regression, either to eliminate the outlier values (like isolation forrest/random forrest) or regresion of higher order. Also, this should be discussed more in the text and underline the main explainatory factors and their possible physiological explanations.

Response 1: Thanks for this helpful comment. We aimed to look at broad trends in the association between environmental factors and seed traits across all six places. We would struggle to justify the exclusion of single data points / places for our broad analysis. We do not think that a more complex statistical analysis would necessarily provide us with better insight and prefer to stick with the broad trends shown.

Point 2: In analysis regarding to meteorological parameters - authors should discuss if such long term view is relevant, or if (for example) effect of last season(s) would show other results. There might be effect of one extreme season (or even extreme month) that needs to be either taken into account or it has to be proven, that there were no such extremes with potential high impact on results (we had simmilar issues with our opium poppy drought stress experiments, even though poppy is one year plant, still there are known seasonal differences even in fruit trees).

Response 2: We agree that there may be differences between the long-term view and short-term extreme weather events that could be masked in the long-term average data. Our challenge though would be how to decide on what constitutes a significant extreme event. This would require the design of a different type of experiment, but prompts us to think that our next study should include the use of real-time data loggers placed at each of the six sites. Having said that, we have taken a look at fruit maturation during the last season and found that temperature and sunshine hours during November provided high correlations with seed DT (r = - 0.59 and - 0.80 respectively). That month coincided with the maturation of the fruit. So, warm sunny days facilitate fruit maturation and that of the seed inside. The related results are now described in the MS. Please see Supplemental Fig.3 and line 161 and 359.

Point 3: Lines 323 - 333 and lines 334-343 are the same, check what should stay.

Response 3: We thank for reviewer’s suggestion. We discussed two different factors on line 326 (seed dry mass) and line 337 (seed initial MC).Thus both comments need to stay in the text, although we have modified the text a little (please see line 337).

Point 4: Line 522 WRKY related genes are not mentioned? Why?

Response 4: Thanks for referee’s suggestion, we added WRKY into the Discussion.

Point 5: Photographies ilustrating differencies in seeds would be useful (even as SM).

Response 5: Thanks for this suggestion. Unfortunately, we didn’t take any photo in the present study, although the variation in seed mass reported really gives a good indication of variability between seeds from the different sites. We plan to take photos in our future studies.

Point 6: Line 475 - should be reformulated - its confusing about total RNA and polyA (mRNA), maybe mRNA enhancement or something like that

Response 6. The sentence has been rewritten, please see line 481.

Point 7:line 478 - reverse transcriptase kit (or enzyme) manufacturer should be stated.

Response 7:  The detail of manufacturer has been added in the MS, please see line 484.

Round 3

Reviewer 2 Report

Authors have responded to all of my comments and the article has been improved seriously and is now ready for publication.

I would sugesst, that parts of the responses would be discussed even in the manuscript (e.g. mention that in future work authors would use...or focus etc.)

Author Response

Point 1: I would sugesst, that parts of the responses would be discussed even in the manuscript (e.g. mention that in future work authors would use...or focus etc.).

Response1: We really appreciated the suggestion from referee, the related discussion was added in the MS, please see line 525.